# Need to Introduce the Finding of Obesity or Normal Body Weight in the Current Diagnostic Criteria and in the Classification of PCOS

**DOI:** 10.3390/diagnostics12102555

**Published:** 2022-10-21

**Authors:** Enrico Carmina

**Affiliations:** Endocrinology Unit, University of Palermo School of Medicine, 90149 Palermo, Italy; enricocarmina28@gmail.com or enricocarmina28@outlook.it

**Keywords:** PCOS, lean PCOS, obese PCOS, PCOS phenotypes, insulin resistance in PCOS

## Abstract

The diagnosis of PCOS is based on the Rotterdam guidelines: chronic anovulation, hyperandrogenism (biologic or clinical) and polycystic ovaries on ultrasound. Two of these three criteria are sufficient for making diagnosis of PCOS. However, one characteristic that is often associated to PCOS (obesity with severe insulin resistance and metabolic alteration regarding glucose metabolism and lipid pattern) has remained out of the current classification of PCOS. Because of this, patients with different metabolic and cardiovascular risk may be included in the same phenotype, and it makes more difficult to establish clear strategies of follow-up and treatment of the patients with increased risks, and also may hide genetic or environmental differences between PCOS patients. Our recent study has shown that metabolic alterations are linked to the weight and not to the Rotterdam phenotypes. Because of this, we suggest a new classification of PCOS phenotypes that divides each Rotterdam phenotype in obese (ob) or lean (l) sub-phenotype. An improved classification of PCOS may be essential for permitting new progress in our understanding of pathogenesis and treatment of PCOS (or of the different disorders that are part of PCOS).

## 1. Introduction

For about 20 years, the diagnosis of PCOS has been based on the Rotterdam guidelines: chronic anovulation, hyperandrogenism (biologic or clinical) and polycystic ovaries on ultrasound [1,2,3]. Two of these three criteria are sufficient for making a diagnosis of PCOS.

The adoption of the Rotterdam criteria has determined the possibility of making a diagnosis of PCOS in patients with very different clinical and biologic characters: patients who have irregular menses and are anovulatory and infertile and fertile ovulatory patients with normal menses, patients who have hyperandrogenism and patients who are normoandrogenic, patients who have polycystic ovaries and patients who have normal ovaries. The possibility of having very different phenotypes has been accepted by the scientific community and, at the moment, four different phenotypes are considered part of this heterogeneous syndrome: A (chronic anovulation, hyperandrogenism and polycystic ovaries; B (chronic anovulation and hyperandrogenism in women with normal ovaries); C (hyperandrogenism and polycystic ovaries in women with regular ovulatory menses); and D (chronic anovulation and polycystic ovaries with normal androgens and clinical signs of hyperandrogenism) [4,5,6,7,8].

However, obesity, a character that is often associated with PCOS, has remained outside of the current classification of PCOS [6]. Because of this, patients with the same PCOS phenotype, according to the Rotterdam criteria, may be lean or severely obese, have very mild (if any) insulin resistance and metabolic alteration, bear an important risk of developing type II diabetes or a normal diabetic risk, having altered lipid pattern with increased cardiovascular risk or normal lipids and no significant cardiovascular risk [6,7,8,9,10]. It represents an important limitation of the actual characterization of PCOS patients. In fact, obesity is associated with severe insulin resistance, metabolic alterations, and cardiovascular complications, and it strongly affects the long-term prognosis of PCOS patients [5]. It also determines the absence of clear strategies for follow-up of these patients, with authors suggesting performing regular controls of the response to oral glucose test [11], or the need for getting a lipid pattern [12] in all PCOS patients, and others suggesting a metabolic evaluation only in a few selected patients [13].

The absence of a role for body weight in the actual classification of PCOS phenotypes may depend on differences in the prevalence of obesity and metabolic alterations between populations [9,10,14,15] but also on the opinion of many experts that obesity and metabolic problems are restricted to patients with phenotype A and B while patients with phenotype C present a mild, non-metabolic form of the syndrome [6,8]. Homogeneous data on patients with phenotype D are missing because in some reports this phenotype is uncommon and with few metabolic problems [10] while in other reports it is common and may present metabolic risks [16]. Differences in studied populations and/or in androgen measurement methods may explain these differences.

## 2. Are the Characters of Obesity and Metabolic Alterations Typical of A and B PCOS Phenotypes?

As previously observed, many authors consider obesity and metabolic alterations a common and largely prevalent character of A and B PCOS phenotypes [6]. However, most studies show completely different data. In many populations, mainly in Mediterranean countries, the Middle East, and East Asia, the majority of PCOS patients have a regular body weight and no or very mild metabolic alterations (mild if any insulin resistance, no increased risk of type II diabetes, normal lipid pattern and no increased cardiovascular risk [9,10,17,18,19,20,21,22]. In addition, metabolic alterations are common also in Phenotype C patients [10]. Also, in US studies, obesity was common in patients with Phenotype C, with no differences in mean body weight between patients with Phenotype C and patients with Phenotype A or B [7].

More importantly, our recent study, part of this special number, shows that in our population (a Mediterranean population) normal body weight is common in all PCOS phenotypes being largely prevalent not only in patients with Phenotype C but also in patients with the classic Phenotype A (and common also in the few patients with Phenotype B) [23]. Only in B phenotype (the only one without polycystic ovaries), that occurs in a minority of patients in all populations, obesity was more common than normal body weight [23].

## 3. Need to Introduce the Finding of Obesity or Normal Body Weight in the Current Classification of PCOS

These findings suggest that each Rotterdam PCOS phenotype includes two very different populations: a lean population that may be anovulatory (Phenotype A or B) or ovulatory (Phenotype C) with little if any metabolic and cardiovascular risk and an obese population who present an increased metabolic and cardiovascular risk independently of their ovulatory status.

Because of this, we suggest introducing into the classification of the PCOS phenotypes the concept of body weight, dividing patients of each phenotype into obese and lean patients (Table 1).

Using the data reported in our recent study on the role of body weight in metabolic risk in different PCOS phenotypes [23], the prevalence of different PCOS phenotypes was calculated (Table 2). As shown in Table 2, phenotypes A-l and C-l (both lean) were the most common, followed by the phenotype A-ob (obese). These data suggest that the differentiation between the obese and lean phenotype is particularly important in the A phenotype (the classic PCOS phenotype that presents chronic anovulation, hyperandrogenism, and polycystic ovaries) because in this phenotype two very common—almost equal in number—populations exist: the first with normal body weight and almost no metabolic risk and the second associated with obesity and important cardiovascular and metabolic risk.

Of course, the prevalence of the different lean or obese phenotypes will be different in different populations but the distinction between obese and lean patients in each Rotterdam phenotype may be particularly useful for establishing clear strategies of metabolic and cardiovascular risk follow up and treatment.

## 4. Open Questions

Many questions remain open.

In our classification we did not include overweight patients. How should we consider these patients, like obese or lean patients?What is the influence of age and treatments on the evolution of body weight? Do obese patients remain obese along the timeline and independently on short term results with diet and physical activity? And what is the cause of the obesity in PCOS?

Certainly, the data are influenced by general characteristics of population, but obesity in PCOS is much more prevalent than in the general population of similar ages and ethnic groups. Is it linked to different a genetic pattern or to environmental factors? Or to a combination of both?

New studies are needed, but an improved classification of PCOS may help to better differentiate PCOS patients. The syndrome is very heterogeneous and may have different causal mechanisms. A better classification of the patients may be essential for permitting further progress in our understanding of pathogenesis and treatment of PCOS (or of the different disorders that are part of PCOS).

## Figures and Tables

**Table 1 diagnostics-12-02555-t001:** Suggested new classification of PCOS. Phenotypes were differentiated according to Rotterdam criteria (A phenotype: chronic anovulation, hyperandrogenism, polycystic ovaries; B phenotype: chronic anovulation, hyperandrogenism but normal ovaries; C phenotype: hyperandrogenism, polycystic ovaries but ovulatory cycles; D phenotype: chronic anovulation, polycystic ovaries but normal androgens) and to finding of obesity (ob = obese) or normal body weight (l = lean).

Phenotype	Chronic Anovulation	Hyperandrogenism	Polycystic Ovaries	Obesity
**A-ob**	+	+	+	+
**A-l**	+	+	+	−
**B-ob**	+	+	−	+
**B-l**	+	+	−	−
**C-ob**	−	+	+	+
**C-l**	−	+	+	−
**D-ob**	+	−	+	+
**D-l**	+	−	+	−

**Table 2 diagnostics-12-02555-t002:** Prevalence of different PCOS phenotypes in a Mediterranean population of PCOS women. Phenotypes were differentiated according to Rotterdam criteria (A phenotype: chronic anovulation, hyperandrogenism, polycystic ovaries; B phenotype: chronic anovulation, hyperandrogenism but normal ovaries; C phenotype: hyperandrogenism, polycystic ovaries but ovulatory cycles; D phenotype: chronic anovulation, polycystic ovaries but normal androgens) and to finding of obesity (ob = obese) or normal body weight (l = lean).

	Prevalence (%)
Phenotype A-ob	**23.9**
Phenotype A-l	**31.2**
Phenotype B-ob	5.9
Phenotype B-l	2.9
Phenotype C-ob	4.4
Phenotype C-l	**29.7**
Phenotype Dob	0.5
Phenotype D-l	1.5

In bold most common PCOS phenotypes according to the suggested new classification [23].

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
