# Peer review of "Need to Introduce the Finding of Obesity or Normal Body Weight in the Current Diagnostic Criteria and in the Classification of PCOS"

_diagnostics, 2022, doi:10.3390/diagnostics12102555_

Round 1

Reviewer 1 Report

To authors,

I believe that the theme is important. The paper is well-written. I have only a small advice. 

I believe that the addition of the following meaning may make the context clearer. 

“Cardiovascular complications and insulin resistance strongly affect the long-term prognosis of PCOS patients: these are strongly associated with current obesity. Thus, presence or absence of obesity has important clinical significance.” 

Anovulation and hyper-androgen may affect fertility = present clinical problem. Morphological change does not directly affect the patient’s health status. Different from these THREE, obesity/lean may “affect” “future health”. But, or therefore, presence/absence of obesity might be important. In short, different from the THREE features, obesity is the problem that poses risks the “future” health. Women do not die due to “anovulation” but they may die due to cardiovascular attack. I mean that “obesity” characterizes different aspects of PCOS, different from the THREE factors. 

I, as a specialist, well understand this point. However, this journal is NOT OBGYN-specific and therefore, shortly adding this meaning might be use for Readers’ better understanding this issue. 

This may not a good example: Every disorder has some types in terms of obesity + or -: for example, liver cirrhosis; obese or not, pre-eclampsia; obese or not. In almost every condition, obese or not might affect the condition and treatment strategies. In PCOS, obesity might have “crucial” meaning, different meaning from live cirrhosis or pre-eclampsia: Your present manuscript does not well describe this context. You need not write long. But, please do consider to touch this issue.  

Please check “parenthesis”. Fundamentally, parenthesis should be avoided if you can express things without it. Especially, “too long” parenthesis should be avoided. You use somewhere wrong parenthesis. Please check and make it correct. 

Author Response

We thank the reviewer for the comments

We understand the importance to explain the role of obesity in affecting long term prognosis of PCOS patients and, because of it, we slightly modified the text and included the comment suggested by the reviewer.

We also modified the tetxt to eliminate a long phrase between parentheses

Reviewer 2 Report

I've read the article on the newly revised PCOS classification with great interest. I believe that the concept presented by the author is exciting and worth discussing.

As PCOS pathology is still under investigation and many issues are unclear, new classifications are significant that would distinguish patient groups from even more data and include additional variables to further develop knowledge in this area to understand the pathology of PCOS better.
It is exciting the proposition of the classification of PCOS phenotypes, the concept of body weight, and the division of patients of any phenotype into obese and slim patients.

Open questions are also worth emphasizing, as they force the reader to reflect and set the directions for further research.

I have no critical remarks and recommend accepting the article in its current form.

Author Response

We thank the reviewer for appreciating our paper